



# Drought propagation and its impact on groundwater hydrology of wetlands

Buruk Kitachew Wossenyeleh[1], Kaleb Asnake Worku[1,2], Boud Verbeiren[2], Marijke Huysmans[1,2]

[1]Department of Earth and Environmental Sciences, KU Leuven University, Leuven, Belgium,
[2]Department of Hydrology and Hydraulic Engineering, Vrije Universiteit Brussel, Brussels, Belgium

*Correspondence to*: Buruk Kitachew Wossenyeleh (burukkitachew.wossenyeleh@kuleuven.be; bkwdae@gmail.com)

## Abstract

Drought can be described as a temporary decrease in water availability over a significant period and affects both surface and groundwater resources. Droughts propagate through the hydrological cycle
and may impact vulnerable ecosystems. This paper investigates drought propagation in the hydrological cycle, focusing on assessing its impact on a groundwater-fed wetland ecosystem. Meteorological drought indices were used to analyze meteorological drought severity. Besides, a method for assessing groundwater drought and its propagation in the aquifer was developed and applied. Groundwater drought was analyzed using the variable threshold method. Furthermore,
meteorological drought and groundwater drought on recharge were compared to investigate drought propagation in the hydrological. This research is carried out in the Doode Bemde wetland in central Belgium.

The results of this research show that droughts are strongly attenuated in the groundwater system. The number and severity of groundwater discharge drought events were smaller than for groundwater
recharge drought. However, the onset of both drought events occurred at the same time, indicating a quick response of the groundwater system to hydrological stresses. In addition, drought propagation in the hydrological cycle indicated that not all meteorological droughts result in groundwater drought. Furthermore, this drought propagation effect was observed in the wetland.

**Keywords:** Groundwater drought, Numerical modelling, Meteorological drought, Variable threshold
method, Wetland



# 1 Introduction

Drought can be described as a temporary decrease in water availability over a significant period of
time. Droughts propagate through the hydrological cycle and affect both surface and groundwater
resources.

If drought concerns groundwater, it is called groundwater drought. Groundwater drought can be
defined as a temporary decrease in groundwater availability over a significant period of time.
Groundwater drought results in decreased groundwater levels and discharges to surface water bodies
(Peters, 2003). As groundwater systems are often slowly responding to drought, groundwater drought
is often characterized by long recovery periods (Calow et al., 1997). Groundwater drought and its
impacts have been less studied than other aspects of drought (Verbeiren et al., 2013; Wilhite and
Glantz, 1985).

In the context of Belgium, Tricot et al. (2015) have shown that drought periods have not intensified
during the last century. The drought periods were defined as the number of consecutive days without
significant precipitation (less than 0.5 mm) for the six hottest months of the year. However, the authors
suggest that their findings do not apply to all forms of drought. For instance, when investigating
groundwater drought, precipitation deficits over longer periods (seasons to years) need to be
investigated (Tricot et al., 2015). Furthermore, climate change projections for Belgium for the coming
century predict drier summers, causing drought conditions (Hoyaux et al., 2016; Tabari et al., 2015).

Groundwater droughts can have severe socio-economic and environmental impacts (Verbeiren et al.,
2013). Streams that are fed by groundwater may run dry if the groundwater system that is feeding
them is affected by drought. These droughts may have significant implications on water supply,
agriculture, and ecology.

Ecosystems such as wetlands that are fed directly by groundwater discharge are also vulnerable to
groundwater drought. Therefore, investigating groundwater drought is especially crucial in nature
reserves such as groundwater-fed wetlands. Wetlands are primarily fed by groundwater and are an
ecologically valuable part of the ecosystem. Droughts such as the one experienced in the summer of



2018 may have a significant impact on the groundwater discharge that is feeding such vulnerable systems. A decrease in groundwater discharge to the wetland may result in a loss of biodiversity. Hence, a thorough investigation of how groundwater discharge is affected by drought is needed.

Groundwater drought can be assessed by investigating groundwater recharge, groundwater level, and groundwater discharge with a high spatial and temporal resolution (Van Lanen and Peters, 2002). Although data on the groundwater levels (H) are typically available with a high temporal resolution, this is usually not the case for groundwater recharge (R) and groundwater discharge (Q) data. Therefore, numerical models are used to simulate these hydrological variables with high temporal and spatial resolution.

Numerical groundwater models are useful to simulating groundwater head and discharge if the necessary data are available and if proper boundary conditions are defined and implemented. Temporal and spatial variations of these variables can be obtained so that further analyses could be performed to identify drought periods. The effect of groundwater abstraction on the occurrence of groundwater drought can also be assessed by incorporating it into the models.  Some previous studies (Peters, 2003; Peters et al., 2006) have used groundwater models to simulate groundwater level and discharge in spatial and temporal scale and investigate groundwater drought using constant threshold value method. In this study, variable threshold value methods and groundwater modelling were combined to investigate groundwater drought in the aquifer.

This paper aims to investigate drought propagation in the hydrological cycle by developing and applying a method for simulating groundwater drought and its propagation in the aquifer with a particular focus on assessing its impact on the hydrological functioning of a groundwater-fed wetland ecosystem. The method combines a water balance model and a groundwater flow model with a threshold method to determine groundwater drought and its effect on groundwater levels and discharge.


## 2 Study site and data

The study area is located in Flanders in Belgium. It is found 8 km to the south of Leuven in the middle
course of the Dijle River (*Fig. 1*). The area contains the nature reserve 'Doode Bemde.' This nature
reserve is an ecologically important wetland. The wetland is primarily fed by groundwater (Verbeiren et
al., 2004).

In the study area, the Dijle river has cut through the surrounding hills to form a 1 km wide and 40 m
deep valley  (De Becker et al., 1999). The hills are situated in the western and eastern parts of the
study area while the valley extends from north to south occupying the middle part of the study area.

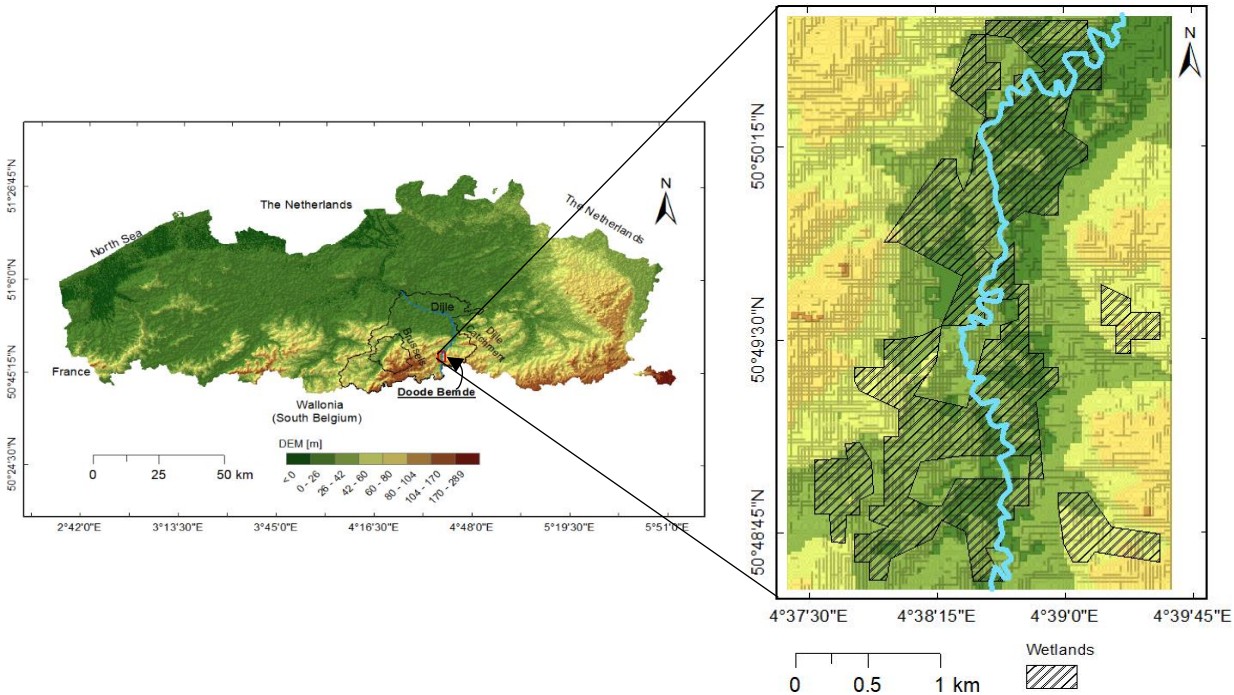

*Figure 1:* Map of Flanders (Belgium) indicating the location of the Doode Bemde nature reserve (Background: relief of Belgium – NGI)

The land cover in the area is predominantly grassland (showing a clear wet-dry gradient) with some
reeds and forest, while few ponds, houses, and streets are also found in the western part of the study
area (Verbeiren et al., 2004). Moreover, around 60% soil in the study area is silty loam and the area
has a gentle slope vary between $0.02^O$ to 11.50º.




Hydrogeological studies on the Dijle catchment show that the top aquifers are unconfined with the thickness ranging from 0 to 50 m. The Dijle river has cut into the sandy tertiary formations of Lede (Ld),

Brussel (Br) and the clayey formation of Kortrijk (Ko). During the Quaternary, a relatively thick layer of loam was deposited in the central valley. A successive deposition of Quaternary deposits near the river further shaped the valley as a result of repetitive inundations (*Fig. 2*).

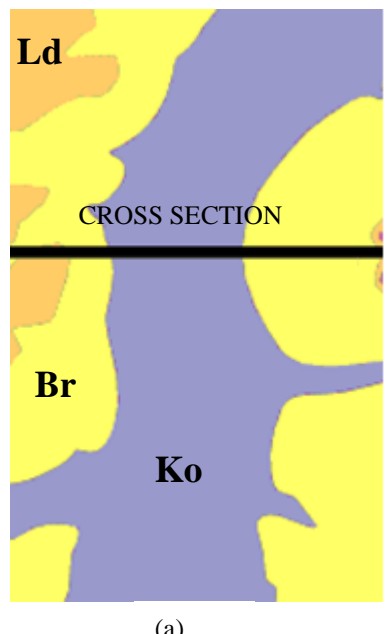

(a)

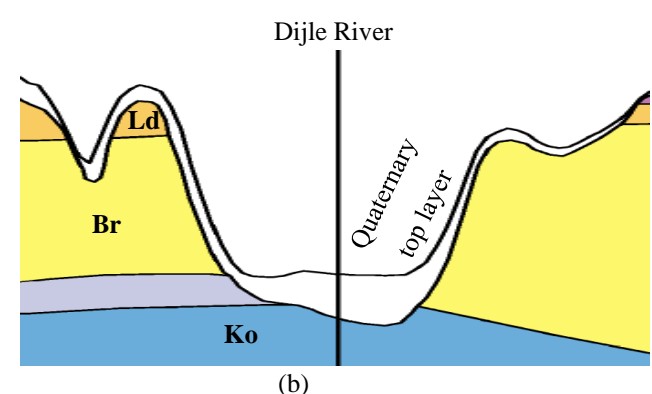

(b)

*Figure 2:* (a) Hydrogeology of Dijle valley around Doode Bemde nature reserve, showing the formations of
Brussel (Ld & Br) and Kortrijk (Ko). (b) A cross section of the shallow hydrogeology (Source:
         Databank Ondergrond Vlaanderen).

Spatially distributed groundwater recharge generated in the GroWaDRISK project funded by BELSPO was used. This project was aiming for the development of a drought-related vulnerability and risk assessment strategy for sustainable management of groundwater resources under temperate

conditions (Verbeiren et al., 2013). In this project, a monthly averaged spatially distributed groundwater recharge was estimated using the WetSpaSS model from 1980 to 2013. WetSpaSS is a physically-based model for the estimation of spatially distributed surface runoff, actual evapotranspiration, and groundwater recharge (Batelaan and De Smedt, 2001). This model accounts for spatially distributed landuse, soil type, slope, elevation, monthly average groundwater depth, and meteorological conditions

as an input (Abdollahi et al., 2017; Batelaan and De Smedt, 2007). However, the recharge obtained

from WetSpaSS was the infiltration at the end of the root zone and did not take in to account the delay of recharge in the unsaturated zone in between the root zone and the groundwater table. Therefore, a delay was implemented on the recharge extracted from the model to account for the time it takes for water to reach the groundwater table from the root zone (Wossenyeleh et al., 2019). The estimated

groundwater recharge is larger than 10 mm/month in most of the study area and spatial average of 22 mm/month (*Fig. 3*). The zones indicated in red have very shallow groundwater levels (abundant water availability) and dense wetland vegetation, resulting in high evapotranspiration rates exceeding infiltration and leading to a "negative" net recharge.

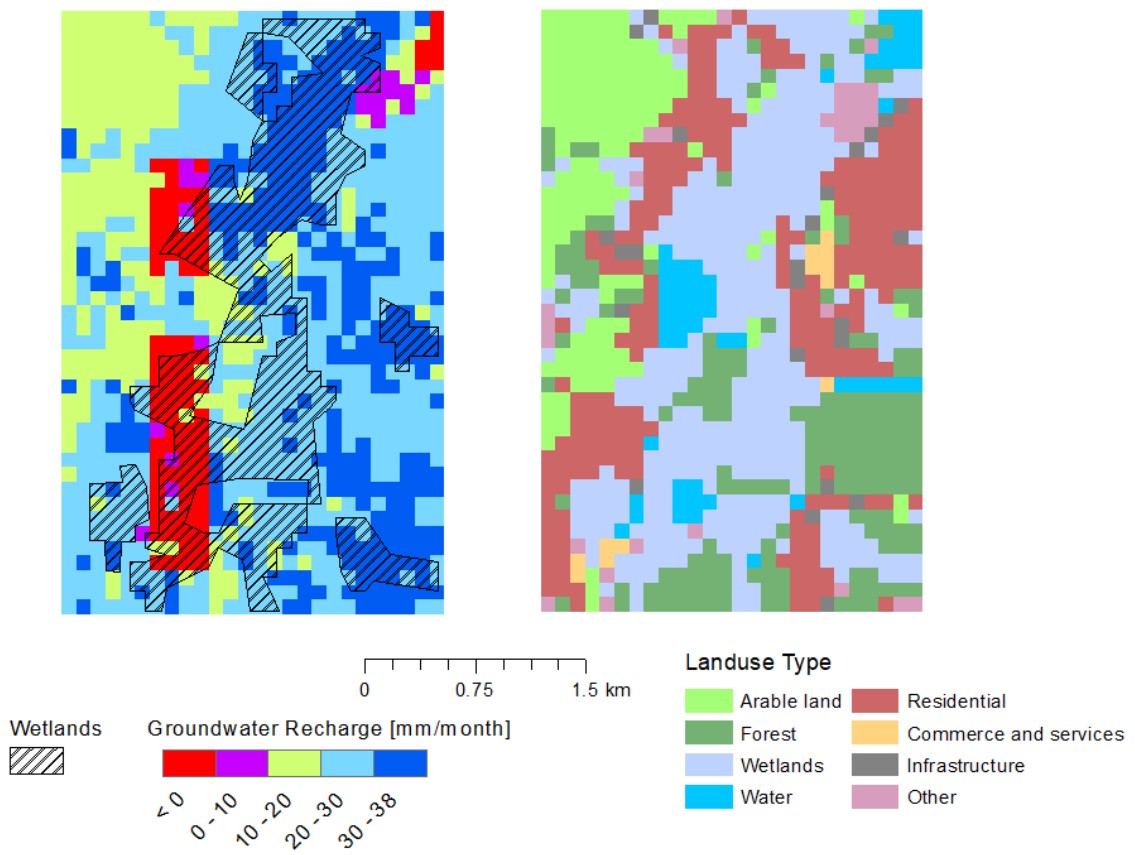

*Figure 3:* Monthly averaged delayed groundwater recharge from 1980 to 2013, estimated using WetSpaSS (right) and land cover class (left)





# 3   Methodology

The study was divided into three parts; meteorological drought analysis, groundwater modelling  and groundwater drought analysis. First, meteorological drought analysis was conducted to investigate
drought on the precipitation. Then, a transient state groundwater model was built to acquire a temporally and spatially distributed simulated groundwater head and discharge. Groundwater drought analysis was performed on simulated groundwater discharge and monthly groundwater recharge time series. Finally, drought analysis was conducted to investigate drought propagation in the hydrological cycle. This was done by comparing groundwater drought to meteorological drought.

## 130   3.1   Meteorological drought analysis

The meteorological drought severity and spatial and temporal extent of the drought were determined using the standardized precipitation index (SPI; McKee et al., 1993). Thirty-one years of monthly averaged precipitation were obtained from the Royal Meteorological Institute (RMI) of Belgium at the Uccle station for the period 1981 to 2011, which was used to calculate SPI and identify meteorological
drought events and severity.

As stated in McKee et al. (1993), SPI is based on the probability distribution of long-term rainfall on different time scales. In this study, the SPI value for 1 month and 12 months were calculated to investigate the long-term meteorological drought severity. Moreover, this drought index was used to analyze the propagation of drought from meteorological to groundwater drought.

The long-term rainfall is fitted to a probability distribution, which is then transformed into a normal distribution so that the mean SPI for the location and desired period is zero (Edward and McKee, 1997). The gamma distribution is used to normalize the rainfall time series (Thom, 1958). Positive SPI values indicate greater than median precipitation, and negative values indicate less than median rainfall. Accordingly, a moderate drought event occurs when the index reaches an intensity of -1.0 or
less (*Table 1*). The event ends when the SPI becomes positive.

According to McKee et al. (1993), drought intensity categories based on the SPI value are defined as shown in *Table 1*.





*Table 1:* Drought intensity categories based on SPI values (McKee et al., 1993)

| SPI Values | Drought category |
| --- | --- |
| 0.0 and more | No |
| 0.0 to -0.99 | Mild |
| -1.0 to -1.49 | Moderate |
| -1.5 to -1.99 | Severe |
| -2.00 and less | Extreme |

## 3.2   Groundwater modelling

Groundwater modelling was performed using the groundwater modelling software MODFLOW-2000 (Harbaugh et al., 2000). A steady state groundwater model for the study area was setup and ran to obtain initial heads for the transient state model, which was built subsequently.

### 155   3.2.1   Conceptual model setup

The conceptual model shown in *Fig. 4* was developed based on information about geology, hydrogeology, and hydrology. The area of interest has an area of 10 km$^2$ and includes most of the Doode Bemde nature reserve and the drinking extraction well operating in the study area.


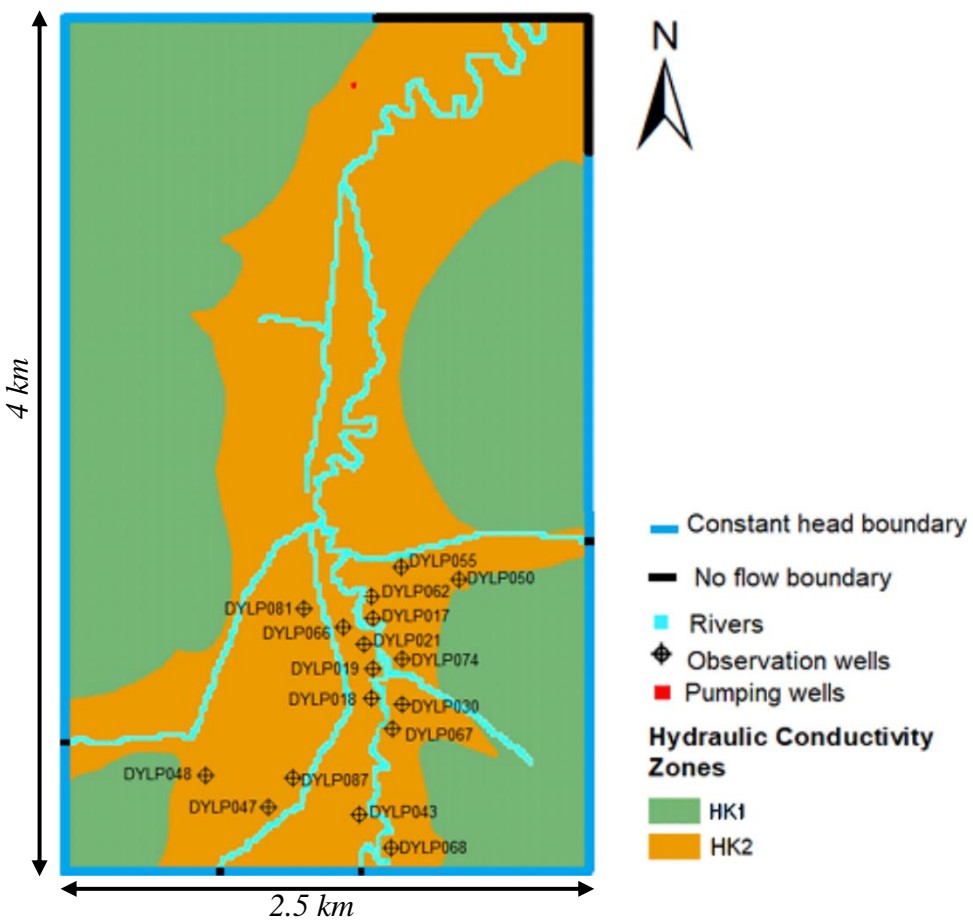

**Figure 4:** Conceptual model setup, showing the specified boundary conditions and hydraulic conductivity zones with two distinct hydraulic conductivities HK1 (Brussels Sand formation) and HK2 (Quaternary loam formation)

The clayey formation of Kortrijk shown in *Fig. 2* was taken as the impervious model bottom due to its much lower hydraulic conductivity compared to the Brussels sand formation and Quaternary loam. Huysmans and Dassargues (2006) reported that the average vertical hydraulic conductivity of the Eocene Ypresian clays, which includes the Kortrijk formation, is 2.1E-8 m/d (with a standard deviation of 4E-8 m$^2$/d$^2$). The Brussels sand formation and Quaternary loam were represented as a one-layer phreatic aquifer system with two hydraulic conductivity zones representing the hill tops and the valley. The hydraulic conductivity of the Brussels sand formation ranges from 0.864 m/d to 43.2 m/d based on the literature review conducted by Vandersteen et al. (2014). Possemiers et al. (2012) also found that the hydraulic conductivity of the Brussels sand is high and has a wide range of reported values





between 2.16 m/d and 63 m/d. An initial value of hydraulic conductivity of 7 m/d for the Brussels sand formation (HK1) was used in this study. Similarly, the hydraulic conductivity range for the Quaternary formation was found to be 1 m/d to 10 m/d (Vandersteen et al., 2014). An initial hydraulic conductivity
of 1 m/d was adopted.

Due to scarcely observed groundwater levels in the vicinity of the study area and no clear natural boundaries (Batelaan et al., 2003), a constant head boundary condition, in which the specified heads were extracted from a regional groundwater model for the Bruland-Krijt groundwater system (VMM, 2008), was implemented for most of the boundary. Moreover, from this regional groundwater model,
the groundwater flow direction is perpendicular to the places where a river enters or exits the study area and where the valley exits the study area. Therefore, no flow boundaries were used for every cell in these places.

For this study, the average river stage and river bottom elevation were interpolated from measurement data obtained at a few locations within the study area (Flemish Water Authorities, 2019). According to
Peeters (2010), most riverbed sediments in the study area consist of Pleistocene sands and gravels. Hence, a high riverbed conductance of 20 m$^2$/d was assumed. Moreover, for every cell that was not modelled as a river within the boundary of the study area, a drain condition was specified. This was done to model the wetland in the study area, which is fed by groundwater adjacent to it and to quantify this groundwater discharge to the wetland from the aquifer. From the study of groundwater flow
modelling of three wetland ecosystems in the river valleys in Flanders by Batelaan et al. (2003) the average drain elevation 0.2 m below topography and a drain conductance of 20 m$^2$/d was assumed in the present study. A groundwater extraction well with an average pumping rate of 900 m$^3$/d is also present in the study area and the model.

To setup and run the model in the transient state, 132 stress periods with monthly time steps from
2003 till 2013 were utilized. Calculated spatially distributed monthly recharge values were used as input for each stress period. Figure 5 shows the seasonal dynamics of groundwater recharge in the study area.




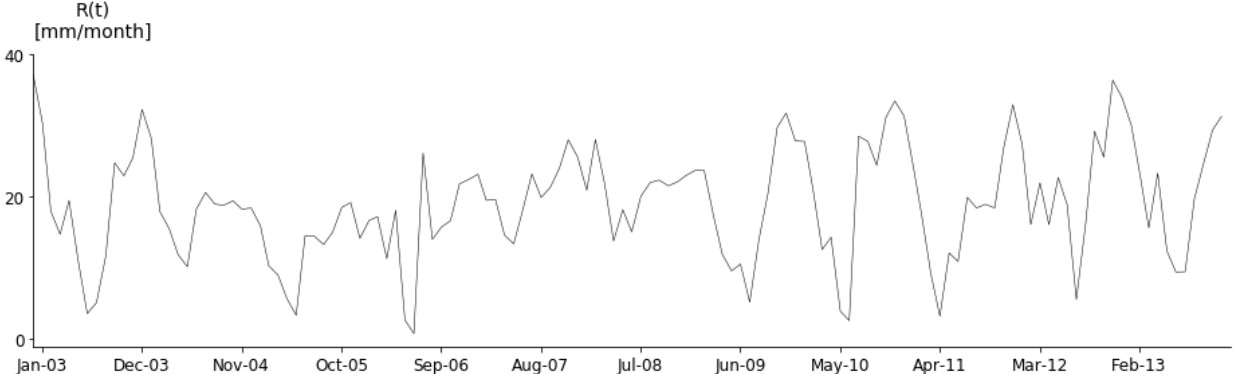

Figure 5: Average monthly groundwater recharge

Relative sensitivities of the model parameters: hydraulic conductivity, drain conductance ($C_{drn}$), specific yield ($S_y$), and river conductance ($C_{riv}$) were done for both Brussels sand formation and Quaternary loam formation to choose calibrated parameter. Average observed hydraulic heads between 2003 and 2013 from 17 observation wells in the study area were used for calibration of the steady state model. 205    Two-weekly observed hydraulic heads from 14 observation wells between 2006 and 2008 were used to calibrate the transient state model. Afterward, the model was validated using observed hydraulic heads between 2011 and 2013. The variance of the differences between observed and simulated heads was used as a quantitative measure of the "success" of the calibration and validation.

### 3.3    Groundwater drought analysis

A groundwater drought analysis was performed on the groundwater recharge (R(t)) and the groundwater discharge (Q(t)) time series of 34 years to investigate the propagation of groundwater drought in the aquifer and its effect on the wetland. The threshold level method introduced by Yevjevich (1967), with a variable threshold value (Beyene and Van Loon, 2014), was used for hydrological drought in deferent geoclimatic conditions. In this study, variable threshold value was used 215    to determine the occurrence of groundwater drought events for both variables.

To do so, first, a separate frequency analysis was conducted for each of the 12 months. The threshold level for each month was determined as the 80th percentile of the probability of exceedance of monthly recharge in that month in the 34-year series (1980-2013). This threshold level is within the 70th-95th percentile of the probability of exceedance range used for most drought studies (Van Loon, 2013). For



each month of the year, the monthly recharge and discharge values in the time series were ranked from highest to lowest. For each month, the 80th percentile of recharge and discharge were

After the separate analysis of drought on groundwater recharge and discharge, the two analyses were combined to characterize groundwater drought events, gain insight into the propagation of drought in the aquifer and assess the response of the groundwater system to drought for the wetland. The

groundwater drought events were investigated in terms of the number of droughts, duration of droughts, and severity (cumulative volume deficit from the threshold level) (Peters, 2003).


## 4   Result and discussion

### 4.1   Meteorological drought

The meteorological drought severity based on monthly SPI (SPI-1) and annual SPI (SPI-12) in the Doode Bemde nature reserve between 1980 and 2011 is shown in Figure 6. As can be seen, there is a difference in severity and frequency between them. Based on SPI-1 value (Fig. 6A), the drought categories vary from extreme to no drought. Whereas SPI-12 value (Fig. 6B) shows less drought severity categories, which varies from severe to no drought.

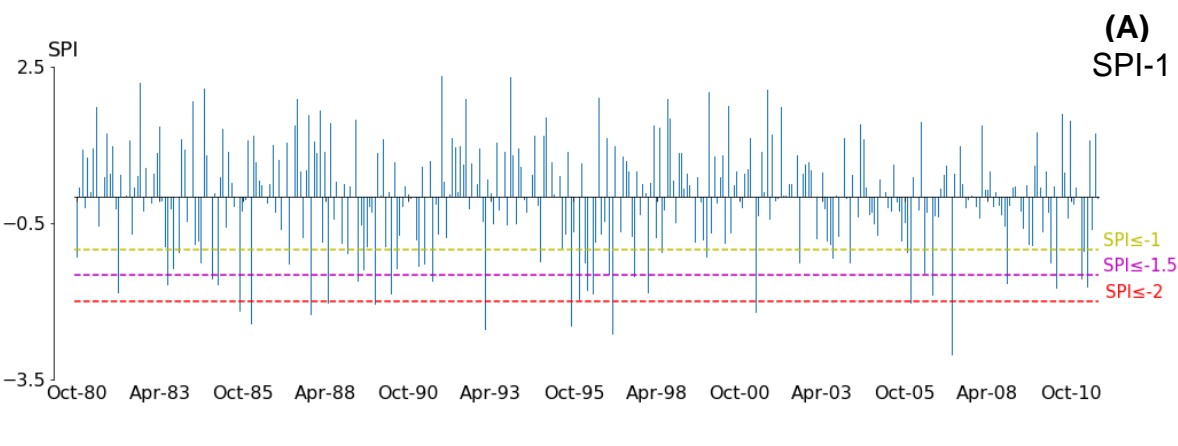

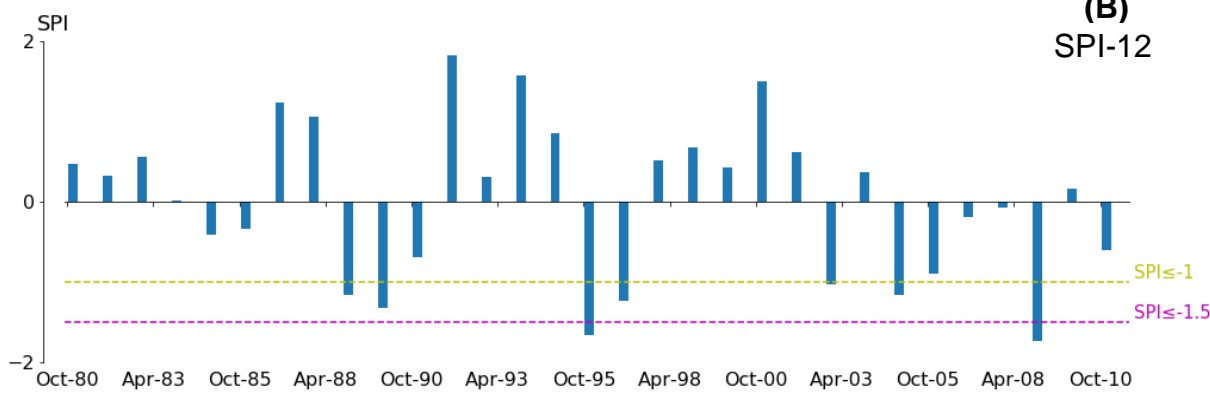


*Figure 6*:  Meteorological drought severity for Doode Bemde nature reserve based on (A) SPI-1 value and (B) SPI-12 value between hydrological year 1980 and 2011

Within the analysis period, eleven extreme meteorological droughts at monthly scale are identified using SPI-1. However, two severe meteorological droughts are identified using SPI-12; these occur in 240   1995-96 and 2008-09. From this result, in a short period SPI calculation, the meteorological drought is frequent but short, and as the period of SPI calculation increases, the duration of drought increase with


less frequency. Therefore, SPI-1 is more convenient for the study of drought propagation in a quick response hydrological system, like in Doode Bemde nature reserve.

## 4.2  Groundwater modelling

Relative sensitivities of model parameters at the sand formation show that hydraulic conductivity of the Brussels sand formation (HK1) was the most sensitive parameter (relative sensitivity of 0.023) followed by drain conductance ($C_{drn}$) (relative sensitivity of 0.009). Specific yield and river conductance had little effect on simulated heads. For the location within the Quaternary loam formation, the river conductance was the most sensitive parameter (0.013) followed by hydraulic conductivity HK2 (0.002)

and drain conductance (0.001). Storage parameters had little effect on calculated heads.

Generally, the relative sensitivities of the model parameters were small, indicating that the uncertainties of the model parameters have limited influence on the simulated heads.

Hydraulic conductivities of the two geologic settings, HK1 for the Brussels sand formation in the hill tops and HK2 for the Quaternary loam in the valley, were optimized during calibration of the steady

state model. After calibration, the variance amounts 0.11 $m^2$. The mean absolute error of the simulated heads was 0.23 m.

The resulting spatial groundwater depth distribution is shown in *Fig. 7* indicated that groundwater went as deep as 44 m in the hills. In the valley, groundwater depths are less than 1 m for the majority of the area but reaches up to 6 m at the western and eastern edges of the valley. Generally, the area has

deep groundwater depths around the hills and shallow groundwater depths around the valley.


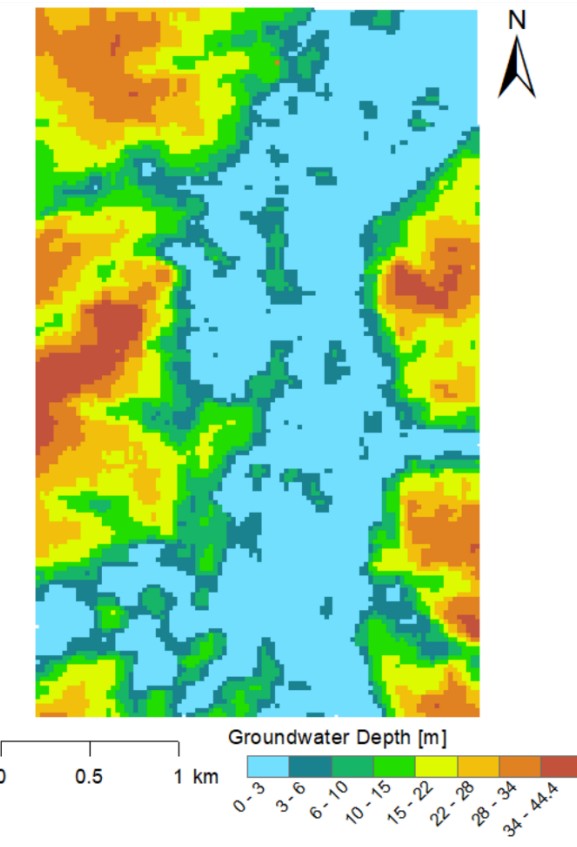

*Figure 7*: Groundwater depth distribution after steady state groundwater model calibration

The steady state groundwater balance was calculated to obtain average fluxes in the study area. Flux across boundaries constituted around 59% of the input into the groundwater system. The remaining input was from recharge. The largest outflow was groundwater discharge to the wetland, represented by drain outflow, which was around 58% of the total output from the aquifer. The aquifer feeds the rivers, and this represents on average around 31% of the total outflow out of the aquifer. The water balance error was 0.01%.

During calibration of the transient model, specific yield was optimized for the two geological settings. The final stage calibration yielded a variance of 0.23 m$^2$ and mean absolute error of 0.39 m between the simulated and observed head. Moreover, the model was validated with observed hydraulic head from a different time period and this yielded a variance of 0.22 m$^2$. The final calibrated values are summarized in Table 2.







*Table 2:* Hydraulic conductivity and Storage parameter value of the two geological settings after calibration

| Parameter | Brussels Sand | Quaternary Loam |
|---|---|---|
| Specific Yield [-] | 0.15 | 0.03 |
| HK [m/d] | 8 | 3 |

Timeseries of observed and simulated head at the observation wells (2-weekly interval) showed that
the model captures the dynamics of groundwater head in the study area reasonably. As example,
*Figure 8* shows the temporal variation of observed and simulated groundwater head at observation well
DYLP081.

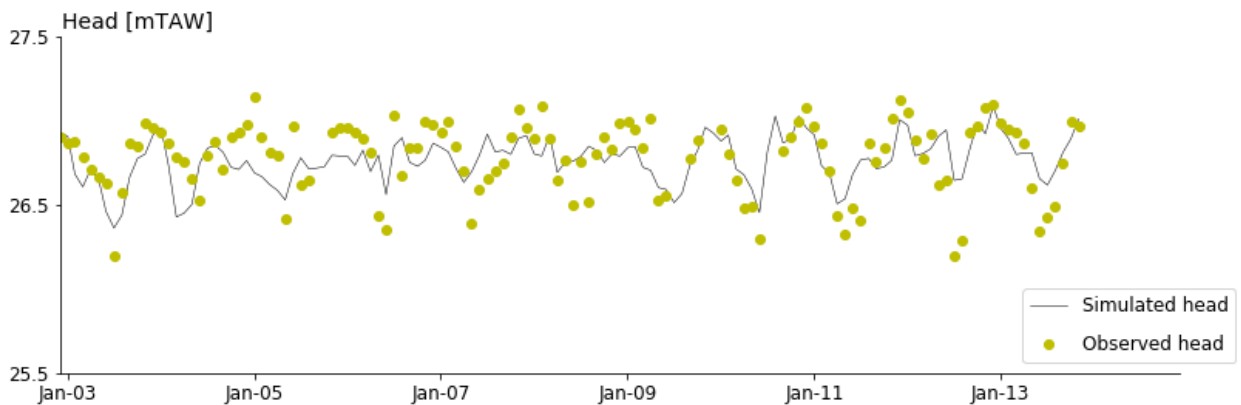

*Figure 8:* Temporarily vary observed and simulated groundwater head at observation well DYLP081   between
285              2003 and 2013

The simulated hydraulic head dynamics between 2003 and 2013 in comparison to groundwater
recharge is shown in *Fig. 9*. The model captured the seasonality of the recharge well. Groundwater
head is high in winter (Dec-Feb) and low in summer (Jun-Aug) There is also no systematic delay in the
timing of the peaks and lows of the hydraulic head and recharge. This could be attributed to shallow
groundwater depths in the study area resulting is a fast response to changes in recharge.

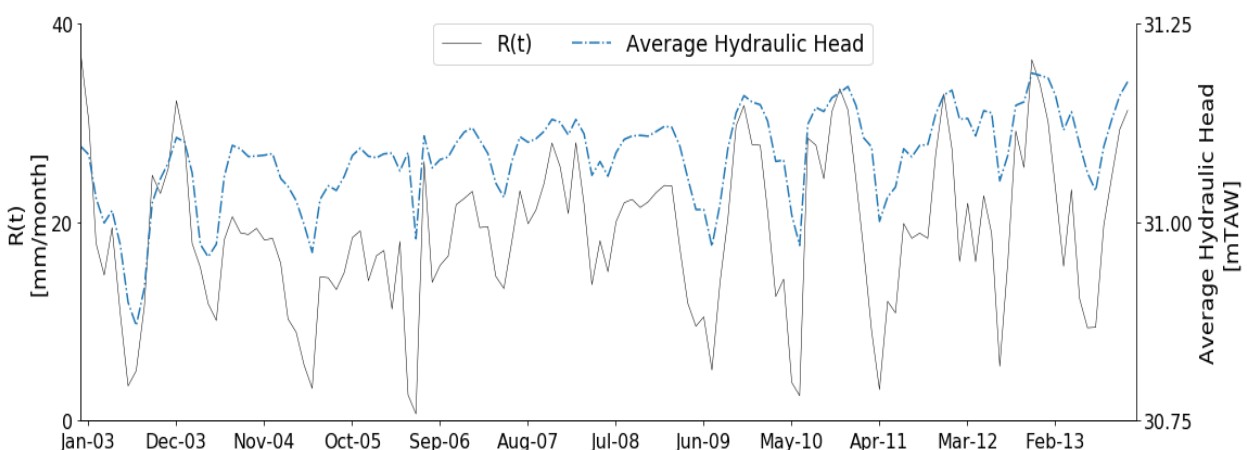

*Figure 9:* Average simulated groundwater head time series and groundwater recharge time series

Simulated total groundwater discharge to the wetland (*Fig. 10*) also shows clear seasonality; high groundwater discharge in winter (Dec- Feb) and low groundwater discharge in summer (Jun-Aug).

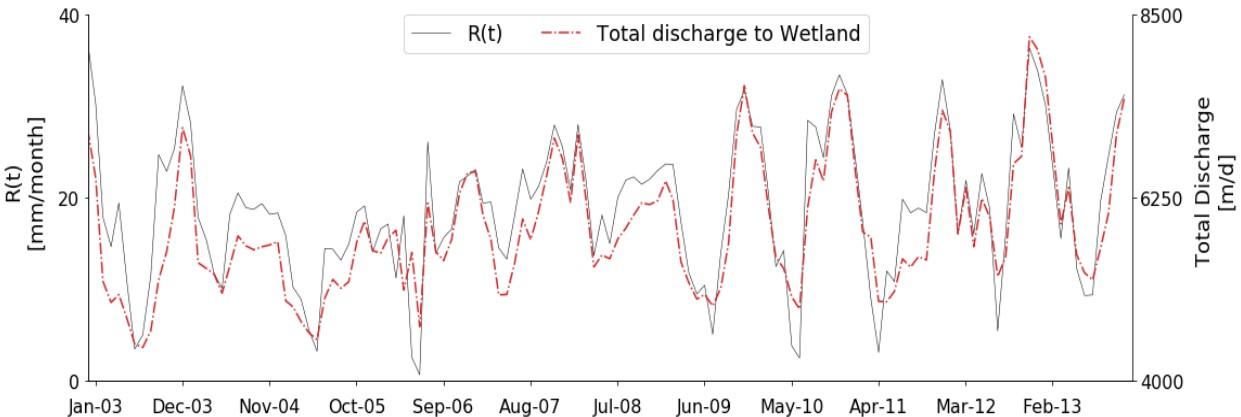


*Figure 10:* Simulated total groundwater discharge and monthly average groundwater recharge

## 4.3 Groundwater drought

### 4.3.1 Groundwater drought on recharge

Groundwater recharge deviation from the threshold and groundwater recharge drought events from 2003 to 2013 are shown in *Fig. 11*. Within this period of analysis, a total of seven groundwater drought events were observed. Minor drought events with cumulative deficit recharge from the threshold of less than 5 mm were ignored. In terms of duration, the drought events had a mean duration of 5 months. The drought with the longest duration was from Oct-04 to Jun-05, which lasted for nine months, closely followed by drought from Sep-05 to Apr-06, which lasted for eight months. Droughts with shorter




durations were observed between Jun-06 and Jul-06 (2 months), between Jun-10 and Jul-10 (2 months) and between Mar-11 and Mar-12 (3 months). The severity of the drought events was assessed based on cumulative deficit recharge from the threshold. The most severe drought was from Oct-04 to Jun-05 with a cumulative deficit recharge of 48 mm, followed by the drought from Sep-05 till Apr-06 with a cumulative deficit recharge of 33 mm. These severe droughts have drought volume higher than the average estimated recharge, which is 22 mm/month.


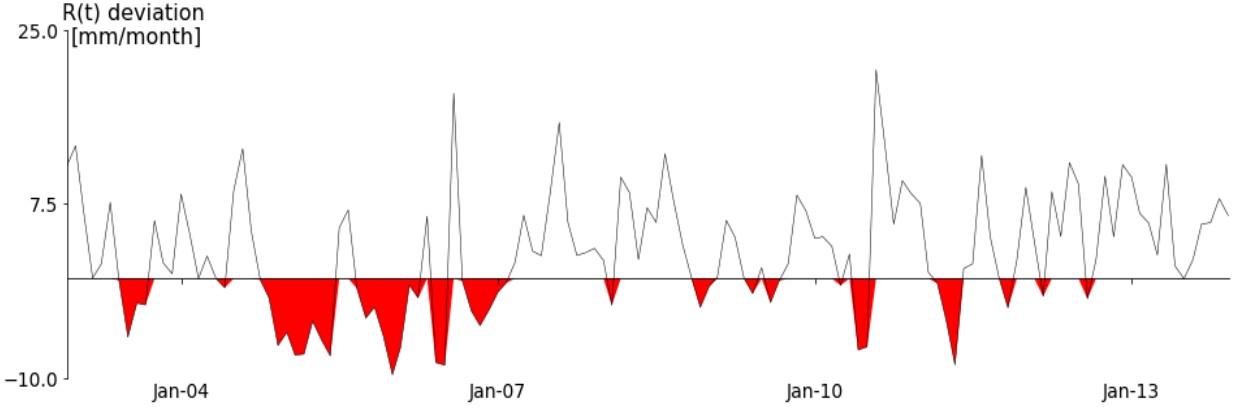

*Figure 11*:Deviation from the threshold value of monthly groundwater recharge. The red color represents groundwater drought events on recharge.

### 4.3.2  Groundwater drought on discharge

Deviations of groundwater discharge from the threshold and discharge drought events are shown in Fig. 12.

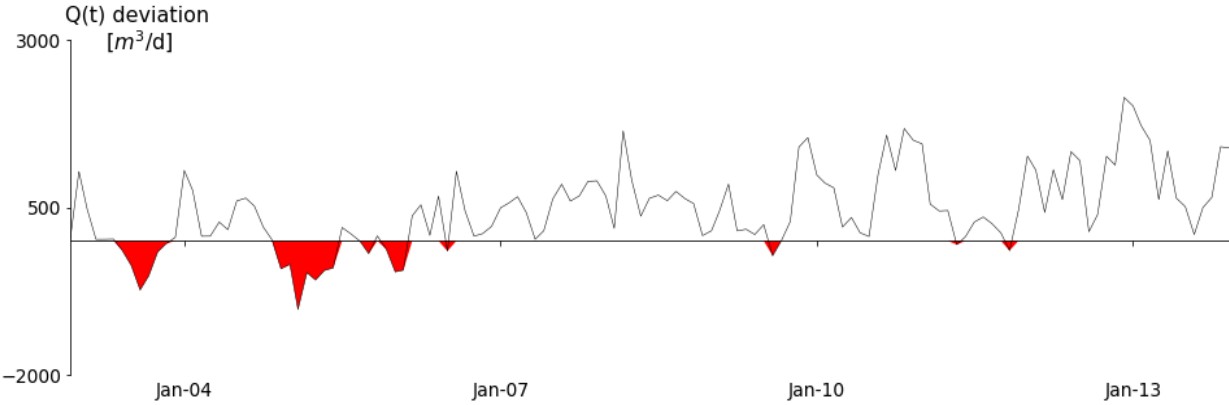

*Figure 12:* Deviation from the threshold value of monthly groundwater discharge to wetland. The red color represents groundwater drought events on discharge.




Three groundwater discharge drought events occurred between 2003 and 2013. The first one was from
Jun-03 till Nov-03. The second drought event occurred from Dec-04 to Jun-05. The last one was from
Dec-05 till Feb-06. Minor drought events with cumulative deficit volume less than 300 m$^3$ were
excluded. The drought events lasted for six months, seven months, and three months respectively,
with a mean duration of 5.3 months. The second drought event from Dec-04 to Jun-05 was the most

severe with a cumulative deficit volume of 3698 m$^3$. The first drought had a cumulative volume deficit
of 1969 m$^3$ and the third 1025 m$^3$.

The average duration of droughts on discharge (5.3 months) showed a slight increase compared to the
mean duration of droughts on recharge (5 months). However, the number of droughts events on
discharge (3) decrease compared to the number of droughts events on recharge (7). This decrement

was expected as the propagation of recharge drought through an aquifer usually results in a lower
number of hydrological droughts (Van Lanen, 2006). Also, recharge droughts were more severe than
discharge droughts (*Fig. 13*). This decrease in severity could be caused by attenuation in discharge
through the groundwater system (Peters, 2003). Another important observation was the absence of
any significant shift in drought events. Approximately, the drought events of recharge and discharge

seem to occur at the same time. This could be attributed to the limited thickness of the aquifer and
shallow groundwater levels in most parts of the study area, which results in a quick response to
hydrologic stresses.

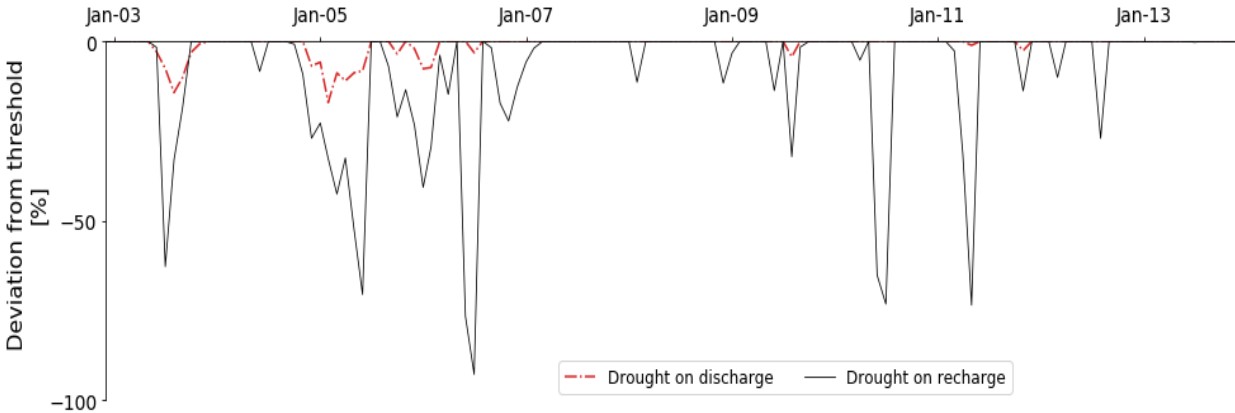

*Figure 13:* Comparison of drought events on groundwater recharge and groundwater discharge using the percent
340              deviation from the threshold



## 4.4 Drought propagation from meteorological to groundwater recharge

The propagation of drought from meteorological to groundwater was analyzed by using SPI-1 and groundwater recharge. Table 3 indicates some of the groundwater drought characteristics between 1981 to 2010 hydrological year. Before every groundwater drought, meteorological drought was
observed in different severity range. Moreover, the severity of the meteorological drought had an impact on the deficit volume and duration of groundwater drought; for example, the groundwater droughts occurred after May-89. In this study area, moderate meteorological drought also propagate to the groundwater, i.e., minor groundwater drought starts at Aug-84. Besides, the lag time between the start of the meteorological drought and the start of the groundwater drought event was calculated. The
lag in Table 3 is the average of the lag of each ground drought events within the major groundwater drought.

Furthermore, by comparing *Fig. 14*(A) and (B), the propagation of drought from meteorological to groundwater recharge can be evaluated. Within the analysis period, meteorological drought is more frequent and observed first than groundwater drought on recharge. In addition, most of the
meteorological droughts propagated to the groundwater recharge with average lag of 9 months, which is attributed to the shallow groundwater depth in most parts of the study area.

*Table 3:* Groundwater drought on recharge characteristics (onset, number events, duration and deficit volume) between 1981 and 2010  and number of meteorological drought (SPI-1) range from moderate to extreme severity before the onset of groundwater drought on recharge with lag time.

| Onset | Duration [month] | Number of drought events | Deficit [mm] | Number of meteorological drought (SPI-1) before onset of groundwater drought on Recharge and lag time [month] |
|---|---|---|---|---|
| Dec-83 | 1 | 1 | 4.95 | 4 (2S, 2Mo) , 14 |
| Aug-84 | 1 | 1 | 2.52 | 1 (1Mo ),1 |
| Oct-85 | 3 | 2 | 2.18 | 4 (1E,2S,1Mo), 9 |
| May-89 | 28 | 10 | 32.97 | 8 (3E,2S, 3Mo),10 |
| Apr-96 | 24 | 6 | 28.63 | 13 (3E,4S,6Mo),11 |
| Jun-03 | 45 | 6 | 123.26 | 9 (2E,3S,4Mo ),18 |
| Feb-08 | 66 | 8 | 45.35 | 8 (2E,1S,4Mo), 6 |

*E: extreme; S: severe; Mo: moderate

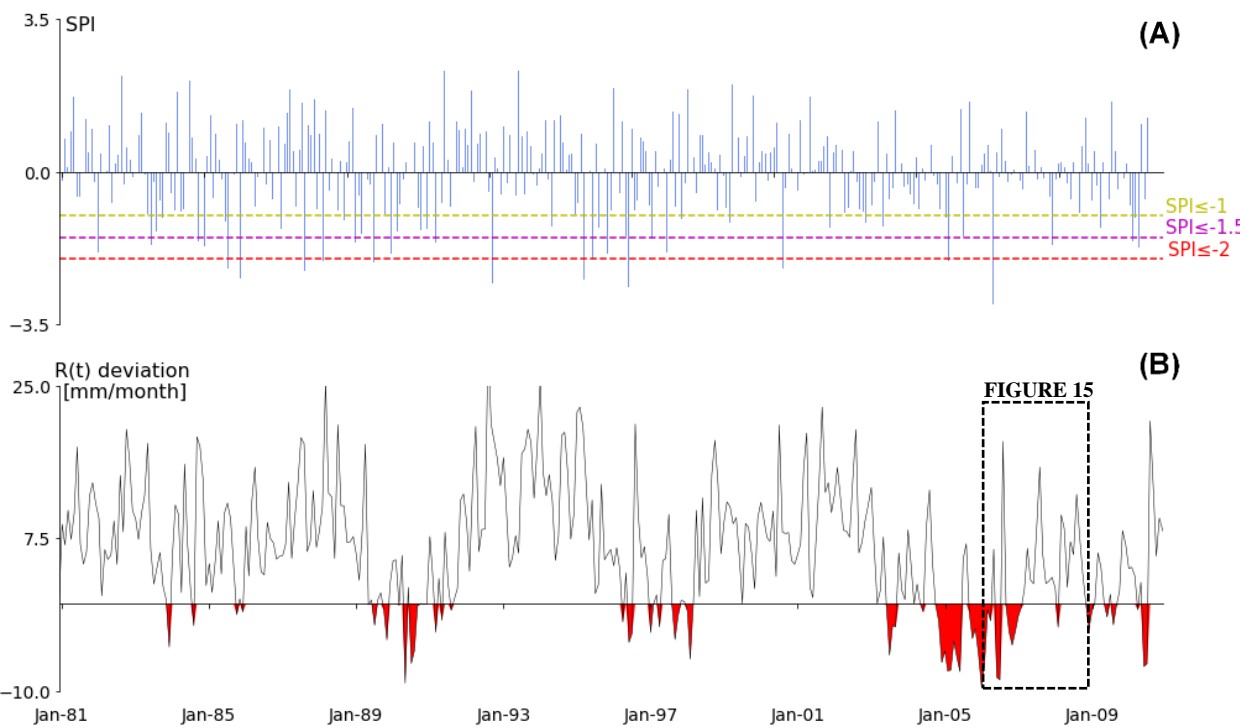

*Figure 14:* Comparison of meteorological drought severity and groundwater recharge drought **(A)** Monthly SPI value and **(B)** Groundwater recharge deviation from the threshold. The red color represents recharge drought events.

365 This propagation of drought had an impact on the hydrological functioning of the groundwater-fed wetland of Doode Bemde. The monthly box plots in *Fig. 15* show observed groundwater head distribution measured using 14 observation wells mostly found on the delineated wetland. Long-term average groundwater heads (green) show clear seasonal dynamics: highest level in January, lowest in June. The mean observed groundwater heads (yellow) are deviating from long term average. The

370 identified groundwater recharge drought (blue) seem to explain the majority of the observed deviation in the period between 2006 to 2008. The lower than long-term average groundwater levels measured in winter (Jan, Feb) and summer (Jun, Jul) of 2006 and winter (Jan, Feb) of 2008 could be the effect of the groundwater droughts that occurred in 2006 and 2008 respectively (Fig. 15). However, the impact of the groundwater drought between fall 2006 and winter 2007 was delayed, which is visible in the

375 groundwater level measurement of the spring of 2007.This delay could be because of the upland recharge from the Dijle valley.



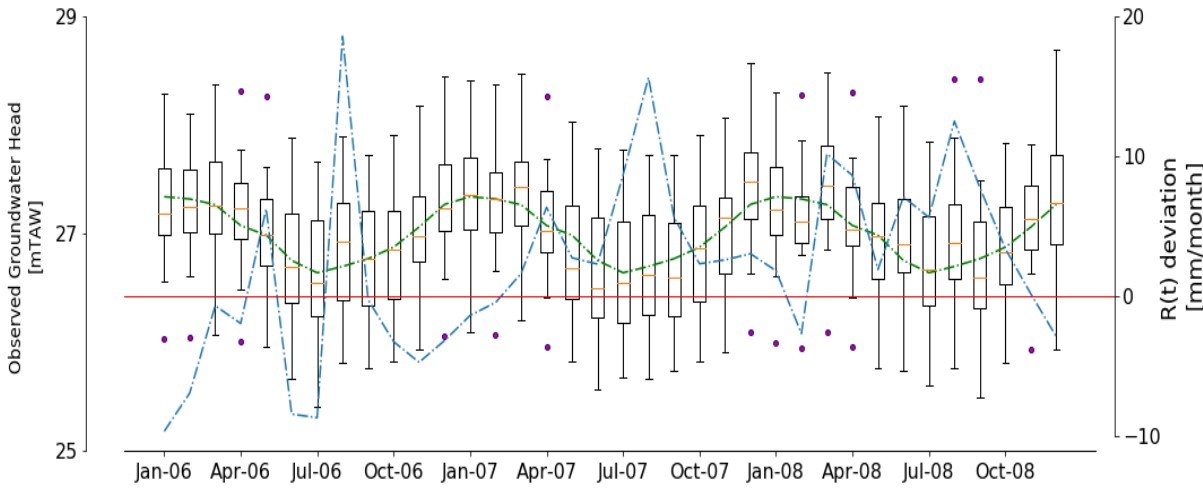

*Figure 15:* Observed groundwater head distribution collected from 14 observation wells (boxplot with monthly timestep), long-term (2003-2013) average monthly observed groundwater head (dotted-green line) and groundwater recharge deviation from threshold value (dotted-blue line) between 2006 and 2008. These observation wells are mainly found on the delineated wetland area (Figure 4).



# 5 Conclusion

In this study, a method is proposed to investigate the propagation of meteorological drought to groundwater recharge, groundwater levels, and groundwater discharge to a wetland. The method combines a water balance model, a transient groundwater flow model, threshold methods, and drought indices. The method is applied to analyze drought propagation in the hydrological cycle, with a focus on groundwater recharge and discharge drought, in the Doode Bemde wetland nature reserve in central Belgium.

Combining water balance model (WetSpass) and groundwater model (MODFLOW) was used to simulate groundwater recharge and groundwater discharge with high temporal and spatial resolution. Moreover, variable threshold value method was implemented for groundwater drought analysis.

Groundwater drought analysis on groundwater recharge and discharge to the wetland showed that the number of groundwater discharge drought events was smaller than the number of recharge drought events. As a result, not all recharge drought events resulted in discharge drought. This was expected as the number of hydrological droughts normally decreases as the groundwater recharge droughts propagate through the aquifer. When it comes to the onset of the drought events, not much difference was observed between droughts on recharge and discharge to the wetland. The drought events appeared to have occurred around the same time. In terms of severity, the drought events on recharge were more severe than the drought events on discharge to the wetland. Attenuation in discharge by the groundwater system could have played a role here.

The groundwater drought analysis showed that even though the number and severity of the drought events on recharge decreased as compared to drought on discharge to the wetland, the wetland is still vulnerable to groundwater drought. This was proven by the lower than long-term average groundwater level measurement during the groundwater drought analysis between 2006 and 2008. This vulnerability could be because of the shallow water table and limited thickness of aquifer in the study area, making it quick to respond to changes in hydrological stresses such as droughts.

From this study, it is concluded that not all meteorological droughts result in groundwater drought. However, a combination of different severity range of meteorological drought causes groundwater drought. Furthermore, the characteristics of drought, like duration, onset, and deficit, changed during



drought propagation in the aquifer and its impact was also detected on Doode Bemde wetland water level.

### *Data Availability statement*

The data that support the findings of this study are available from the corresponding author upon
415    reasonable request.

### *Author Contributions*

Led the development of this work: B.K.W.; Conceptualization, B.K.W., K.A.W.,  B.V., and M.H.; data collection, B.K.W., K.A.W., B.V., and M.H.; methodology, B.K.W., K.A.W., B.V., and M.H.; supervision, M.H. and B.V.; writing -original draft, B.K.W., and K.A.W.; and writing - review and editing, B.K.W.,
420    B.V., and M.H. All authors have read and agreed to the published version of the manuscript.

### *Acknowledgments*

This research was supported by a KULeuven interfaculty council for development cooperation (IRO) PhD scholarship. The research idea, and spatially distributed groundwater recharge time series were
425    obtained from the GRoWaDRISK project (SD/RI/05a, 2012-2016), funded by the Belgian Science Policy Office (BELSPO) within the framework of the 'Science for Sustainable Development' research programme.



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
