# Peer review of "Drought propagation and its impact on groundwater hydrology of wetlands: A case study on the Doode Bemde nature reserve (Belgium)."

_Natural Hazards and Earth System Sciences, 2020_

## Referee Comment (RC1) · Anonymous Referee #1 · 17 Aug 2020

**General comments:**

Dear authors, the paper is very interesting, and it is appreciated that in general is well-structured and written. I enjoy reading it! However, there is still room for improvement in the language and some typos. Please be careful in how the type of drought and its impacts is written through the text. Since sometimes these concepts are mixed up.

The methodology applied is well explained. Even though the results and discussion chapter should include more balanced discussions with other authors, concepts and related/similar work. Please add it.

More specific comments and technical questions and clarifications are given in the lines below:

**Specific comments:**

P1, L14- 16. Please rewrite the sentence; it is hard to read.

P1, L20. Change or delete the word drought after the groundwater recharge

P2, L28-29 and L32. Add references

P2, 41. Please specify for which type(s) of drought(s) the author made the assessment

P3, 53. Add reference, for the readers would be interesting to know more about the 2018 drought and its implications.

P3, L69-73 It should be stated here that also meteorological indicators are part of the assessment

P5, L102. Please introduce the acronym GroWaDRISK and the other acronyms.

P7, L137. Why are you using SPI 1 if you are only focusing in long term meteorological drought? Please clarify.

P10, L172. Why did you choose an "initial value of hydraulic conductivity of 7 m/d for the Brussels sand formation" despite the wide range? Please also clarify on the text.

P11, L202. Specific yield needs to be introduced as the other parameters before.

P11, L211. Specify the year period you used.

P12, L221. The sentence is missing.

P13, L230. Is missing the discussion with other authors about the differences when using SPI-1 or 12 for drought assessments.

P16, L297. In the chapter Groundwater drought. Why did you exclude minor drought events in each of the subchapter analysis? How do you select the threshold to ignore those minor drought events for each subchapter?

P19, L331-332. Please, reconsider changing this sentence to "drought events on groundwater recharge were more severe than groundwater discharge". As you are talking about drought impacts on groundwater discharge and recharge.

P20, L341. Why are you showing results here until 2010, if your assessment was performed until 2011?

P23, L402-408. This paragraph fits better on the discussion.

**Technical corrections:**

P9, L168. Hilltops is miswritten, please correct accordingly

P10, L181. I suggest adding a comma after "enters or exits the study area" to make it easier to read.

P11, L214. Correct the typo "deferent"

P13, L237. A preposition is missing

P16, L281. "Figure 8" is in italic

P16, L288. Missing point before "There is"

P16, L290. Typo in the sentence "resulting is a fast response"

---

## Editor Comment (EC1) · Ana Iglesias (Editor) · 24 Sep 2020

Very interesting and well written.

A few comments:

1. Title. Since the study focuses in a case study with very particular hydro-geological conditions, the name of the case study may be included in the title.

2. Abstract: The last paragraph of the abstract is confusing and seems contradictory.

3. Methods: Add a section on validation of the simulation model with empirical data, and an example by using the time series of Figure 5 (200 Average monthly groundwater recharge)

---

## Referee Comment (RC2) · Ana Iglesias (Referee) · 26 Oct 2020

The manuscript is very interesting and well written, proposing a method to analyse the the propagation of meteorological drought to groundwater recharge, groundwater levels, and groundwater discharge to a wetland. Some changes could make the manuscript clearer and more focused, especially in the Methods section. Below are some comments to the authors:

1. Title. Since the study focuses in a case study with very particular hydro-geological conditions, the name of the case study may be included in the title.

2. Abstract: The last paragraph of the abstract is confusing and seems contradictory.

3. paragraph 30. Please revise to include meteorological drought.

[Figure]

4. paragraph 40. First line, what type of droughts? The results in Figure 5 disagree with this statement. This result should be highlighted.

5. Methods: Add a section on validation of the simulation model with empirical data, and an example by using the time series of Figure 5 (200 Average monthly groundwater recharge).

6. Methods: Add a framework figure, linking the modelling components.

7. Results and discussion: As written now, it is a summary of individual results of the different modelling components. The manuscript will benefit from a more comprehensive text that links the results to the final goal of analysis.

---

## Author Response (AR1)

*Reviewer #1*

Dear Reviewer,

We appreciate the time and effort that you dedicated to read our manuscript and for the valuable comments, you have provided. We answer your comments, suggestions and questions below and have incorporated them into the manuscript. Your constructive comments have significantly improved the quality and clarity of our manuscript.

Please find below our point by point response to each of the comments.

Best regards,

Buruk Kitachew Wossenyeleh
On behalf of all authors

Part I: General Comments:

*Point 1:* Dear authors, the paper is very interesting, and it is appreciated that in general is well-structured and written. I enjoy reading it! However, there is still room for improvement in the language and some typos. Please be careful in how the type of drought and its impacts is written through the text. Since sometimes these concepts are mixed up. The methodology applied is well explained. Even though the results and discussion chapter should include more balanced discussions with other authors, concepts and related/similar work. Please add it.

**Response 1:** Thank you for the positive feedback of the paper and your recommendation. We agree discussing the results of other authors would add value to the manuscript and have added some discussion about the suggested point. We also encourage similar studies to be done in other regions. We included the following discussion in the result and discussion part of the revised manuscript.

Line [386-398] During drought propagation in the hydrological cycle, the multi-year meteorological droughts of 1981 – 2013 propagate to groundwater recharge drought. This propagation continues to the groundwater system. The groundwater drought propagation analysis showed that even though the number and severity of drought events observed in discharge to the wetland is lower than for recharge, , the wetland is still vulnerable to groundwater drought. This is also reflected in  the lower groundwater level measurements between 2006 and 2008. This

vulnerability could be because of the shallow water table and limited thickness of the aquifer in the study area, resulting in a quick response to changes in hydrological stresses such as droughts. The drought propagation towards a wetland studied by Fang and Pomeroy (2008) also showed much lower discharge to the wetland from the basin groundwater and snowmelt runoff developed in drought years. Moreover, Drexler and Ewel (2001) performed a field experiment during the 1997–1998 ENSO-related drought and found that the mean water table level in the wetlands lowered by 12 to 54 cm. This could also be explained by drought propagation from the meteorological to the groundwater system.

Part II: Specific comments:

*Point 1:* P1, L14- 16. Please rewrite the sentence; it is hard to read.

**Response 1:** Agree and changes made in the revised manuscript.

Line [15-17] Furthermore, meteorological drought and groundwater drought on recharge were compared to investigate drought propagation in the hydrological cycle.

*Point 2:* P1, L20. Change or delete the word drought after the groundwater recharge

**Response 2:** Agree and changes made. We included the following rewritten sentence in the revised manuscript.

Line [19-20] The number and severity of drought events on groundwater discharge events were smaller than for groundwater recharge.

*Point 3:* P2, L28-29 and L32. Add references

**Response 3:** Agree and changes made. We included the references in the revised manuscript.

Line [29-32] Drought can be described as a temporary decrease in water availability over a significant period and caused by deficient precipitation. Droughts propagate through the hydrological cycle and affect both surface and groundwater resources (Bloomfield and Marchant, 2013; Calow et al., 1997; Mishra and Singh, 2010; Wilhite, 2000).

***Point 4:*** P2, 41. Please specify for which type(s) of drought(s) the author made the assessment

**Response 4:** Agree and changes made. We included the type of drought in the revised manuscript.

Line [41-42] The meteorological drought periods were defined as the number of consecutive days without significant precipitation (less than 0.5 mm) for the six hottest months of the year.

***Point 5:*** P3, 53. Add reference, for the readers would be interesting to know more about the 2018 drought and its implications.

**Response 5:** Agree and changes made. We included the references in the revised manuscript.

Line [55-57] Droughts such as the one experienced in the summer of 2018 may have a significant impact on the groundwater discharge that is feeding such vulnerable systems (Ridder et al., 2020).

***Point 6:*** P3, L69-73 It should be stated here that also meteorological indicators are part of the assessment

**Response 6:** Agree and we revised the aim of the paper. We included the meteorological drought in the aim of the paper as follows:

Line [75-77] This paper aims to investigate drought propagation in the hydrological cycle by developing and applying a method for simulating meteorological drought, groundwater drought, and its propagation in the aquifer, particularly focusing on assessing its impact on groundwater-fed wetland ecosystem.

***Point 7:*** Please introduce the acronym GroWaDRISK and the other acronyms.

**Response 7:** Agree and changes made.

Line [104-106] Spatially distributed groundwater recharge generated in the GroWaDRISK (Drought-related vulnerability and risk assessment of groundwater resources in Belgium) project funded by Belgian Federal Science Policy Office (BELSPO) was used.

***Point 8*** P7, L137. Why are you using SPI 1 if you are only focusing in long term meteorological drought? Please clarify.

**Response 8:** In our study, meteorological drought was analyzed using SPI 1 and SPI 12. Moreover, the study also investigates drought propagation from meteorological drought to groundwater

drought. Because of the hydrological response of the study area, SPI-1 is more convenient for the study of drought propagation. It helps to see the short term drought propagation in hydrological cycle.

This explanation was stated in the manuscript as:

Line [247-248] Therefore, SPI-1 is more convenient for the study of drought propagation in a quickly responding hydrological system, like in the Doode Bemde nature reserve.

**Point 9:** P10, L172. Why did you choose an "initial value of hydraulic conductivity of 7 m/d for the Brussels sand formation" despite the wide range? Please also clarify on the text.

**Response 9:** Agree and changes made. This initial hydraulic conductivity is selected from the hydraulic conductivity ranges found by different researchers. This value is optimized during the calibration of the model. The optimized parameter value is stated in the results part of the manuscript. We included these explanations in the revised as follows:

Line [176-180] Therefore, from these ranges, an initial value of hydraulic conductivity of 7 m/d for the Brussels sand formation (HK1) was used in this study. Similarly, the hydraulic conductivity range for the Quaternary formation was found to be 1 m/d to 10 m/d (Vandersteen et al., 2014). An initial hydraulic conductivity of 1 m/d was adopted. These initial values of hydraulic conductivities are optimized during the calibration of the model.

**Point 10:** P11, L202. Specific yield needs to be introduced as the other parameters before.

**Response 10:** Agree. It is already introduced in the model as well as in the manuscript.

Line [205-207] Relative sensitivities of the model parameters: hydraulic conductivity, drain conductance (Cdrn), **specific yield (Sy)**, and river conductance (Criv) were executed for the Brussels sand formation and Quaternary loam formation to select parameters for calibration.

**Point 11:** P11, L211. Specify the year period you used.

**Response 11:** Agree and changes made.

Line [215-217] A groundwater drought analysis was performed on the groundwater recharge (R(t)) and the groundwater discharge (Q(t)) time series of 34 years (1980-2013) to investigate the propagation of groundwater drought in the aquifer and its effect on the wetland.

**Point 12:** P12, L221. The sentence is missing

**Response 12:** Agree and changes made.

Line [226] For each month, the 80th percentile of recharge and discharge were calculated.

**Point 13:** P13, L230. Is missing the discussion with other authors about the differences when using SPI-1 or 12 for drought assessments.

**Response 13:** Agree and changes made. Our aim in this part of the analysis is to choose convenient SPI accumulation period for drought propagation from meteorological to groundwater. Based on your recommendation we include the following sentence in the revised manuscript.

Line [248 - 250] Cammalleri and Barbosa (2019) also showed a short period SIP calculation is suitable for a quickly responding hydrological system.

**Point 14:** P16, L297. In the chapter Groundwater drought. Why did you exclude minor drought events in each of the subchapter analysis? How do you select the threshold to ignore those minor drought events for each subchapter?

**Response 14**: Agree and changes made. All drought events are included in the analysis of the subchapter. The sentence is rewritten in the revised manuscript as follows:

Line [303-305] Groundwater recharge deviation from the threshold and groundwater recharge drought events from 2003 to 2013 are shown in Fig. 12. Within this analysis period, seven drought events with a severity higher than 5 mm of cumulative deficit recharge were observed.

**Point 15:** P19, L331-332. Please, reconsider changing this sentence to "drought events on groundwater recharge were more severe than groundwater discharge". As you are talking about drought impacts on groundwater discharge and recharge.

**Response 15:** Agree and changes made.

Line [335-336]: drought events on recharge were more severe than groundwater discharge.

**Point 16:** P20, L341. Why are you showing results here until 2010, if your assessment was performed until 2011?

**Response 16:** Agree and changes made. We change 2010 to 2011 in the revised manuscript.

**Point 17:** P23, L402-408. This paragraph fits better on the discussion.

**Response 17:** Agree and changes made. This conclusion part moved to the discussion part in the revised manuscript.

Part III: Technical corrections:

**Response Part III:** Agree and all corrections are made.

*Reviewer #2*

Dear Ana Iglesias,

We appreciate the time and effort that you dedicated to read our manuscript and for the valuable comments, you have provided. We answer your comments, suggestions and questions below and have incorporated them into the manuscript. Your constructive comments have significantly improved the quality and clarity of our manuscript.

Please find below our point by point response to each of the comments.

Best regards,

Buruk Kitachew Wossenyeleh
On behalf of all authors

Part I: General Comments:

**Point 1:** The manuscript is very interesting and well written, proposing a method to analyses the propagation of meteorological drought to groundwater recharge, groundwater levels, and groundwater discharge to a wetland. Some changes could make the manuscript clearer and more focused, especially in the Methods section.

**Response 1:** Thank you for the positive feedback of the paper and your recommendation.

Part II: Specific comments:

**Point 1:** Title. Since the study focuses in a case study with very particular hydro-geological conditions, the name of the case study may be included in the title.

**Response 1:** Agree and changes made. We included the case study are in the title as follows:

Title: Drought propagation and its impact on groundwater hydrology of wetlands: A case study on the Doode Bemde nature reserve (Belgium).

**Point 2:** Abstract: The last paragraph of the abstract is confusing and seems contradictory.

**Response 2:** In the abstract, we want to show how the drought changes during the propagation in the groundwater system. The drought is attenuated (reduced its severity and event number) when it propagates through the hydrological cycle. We rewrote the paragraph in the revised manuscript to make it more clear.

Line [19-23] The results of this research show that droughts are attenuated in the groundwater system. The number and severity of drought events on groundwater discharge were smaller than for groundwater recharge. However, the onset of both drought events occurred at the same time, indicating a quick response of the groundwater system to hydrological stresses. In addition, drought propagation in the hydrological cycle indicated that not all meteorological droughts result in groundwater drought.

**Point 3:** paragraph 30. Please revise to include meteorological drought.

**Response 3:** Agree and changes made in the revised manuscript.

Line [29-32] Drought can be described as a temporary decrease in water availability over a significant period and caused by deficient precipitation. Droughts propagate through the hydrological cycle and affect both surface and groundwater resources (Bloomfield and Marchant, 2013; Calow et al., 1997; Mishra and Singh, 2010; Wilhite, 2000).

**Point 4:** paragraph 40. First line, what type of droughts? The results in Figure 5 disagree with this statement. This result should be highlighted.

**Response 4:** Agree and changes made. We included the type of drought in the revised manuscript.

The result in Figure 5 shows the time series distribution of a monthly average groundwater recharge. The statement by Tricot et al. (2015) was about meteorological drought while the figure shows recharge, which might explain the confusion. We added the following explanation to clarify this:

Line [41-42] The meteorological drought periods were defined as the number of consecutive days without significant precipitation (less than 0.5 mm) for the six hottest months of the year.

**Point 5:** Methods: Add a section on validation of the simulation model with empirical data, and an example by using the time series of Figure 5 (200 Average monthly groundwater recharge).

**Response 5:** No change made because we believe this is already discussed in the current form of the manuscript.

Explanation:

How the model performed during the validation period was shown in the Figures 9, 10, and 11 (Figures 8, 9, and 10 in the old version). These figures include both the calibration period (2006 – 2008) and the validation period (2011- 2013) stated in the manuscript. Therefore, the model simulation result was compared with the empirical data (observed groundwater head and groundwater recharge) during the validation period.

**Point 6:** Methods: Add a framework figure, linking the modelling components.

**Response 6:** Agree and changes made. We include the following conceptual framework of the study in the revised manuscript.

[Figure]

Line [129] Figure 4: Conceptual framework of the study

**Point 7:** Results and discussion: As written now, it is a summary of individual results of the different modelling components. The manuscript will benefit from a more comprehensive text that links the results to the final goal of analysis.

**Response 7:** Agree and we included the following discussion in the result and discussion part of the revised manuscript.

Line [386-398] During drought propagation in the hydrological cycle, the multi-year meteorological droughts of 1981 – 2013 propagate to groundwater recharge drought. This propagation continues to the groundwater system. The groundwater drought propagation analysis showed that even though the number and severity of drought events observed in discharge to the wetland is lower than for recharge, , the wetland is still vulnerable to groundwater drought. This is also reflected in  the lower groundwater level measurements between 2006 and 2008. This vulnerability could be because of the shallow water table and limited thickness of the aquifer in the study area, resulting in a quick response to changes in hydrological stresses such as droughts. The drought propagation towards  a wetland studied by Fang and Pomeroy (2008) also showed much lower discharge to the wetland from the basin groundwater and snowmelt runoff developed in drought years. Moreover, Drexler and Ewel (2001) performed a field experiment during the 1997–1998 ENSO-related drought and found that the mean water table level in the wetlands lowered by 12 to 54 cm. This could also be explained by drought propagation from the meteorological to the groundwater system.

***Marked-up manuscript version***

[revised manuscript text omitted]

During drought propagation in the hydrological cycle, the multi-year meteorological droughts of 1981 – 2013 propagate to groundwater recharge drought. This propagation continues to the groundwater system. The groundwater drought propagation analysis showed that even though the number and severity of drought events observed in discharge to the wetland is lower than for recharge, , the wetland is still vulnerable to groundwater drought. This is also reflected in the lower groundwater level measurements between 2006 and 2008. This vulnerability could be because of the shallow water table and limited thickness of the aquifer in the study area, resulting in a quick response to changes in hydrological stresses such as droughts. The drought propagation towards a wetland studied by Fang and Pomeroy (2008) also showed much lower

discharge to the wetland from the basin groundwater and snowmelt runoff developed in drought years. Moreover, Drexler and Ewel (2001) performed a field experiment during the 1997–1998 ENSO-related drought and found that the mean water table level in the wetlands lowered by 12 to 54 cm. This could also be explained by drought propagation from the meteorological to the groundwater system.

**5   Conclusion**

[revised manuscript text omitted]

VMM: Grondwater in Vlaanderen: Het Brulandkrijtsysteem., Aalst, Belgium, 2008.

Wilhite, D.A.: Chapter1 Drought as a Natural Hazard: Concepts and Definitions, Drought A Glob. Assess. 1, 3–18, 2000.

Wilhite, D. A. and Glantz, M. H.: Understanding the drought phenomenon: The role of definitions, Water Int., 10, 111–120, doi:10.4324/9780429301735-2, 1985.

Wossenyeleh, B.K., Verbeiren, B., Diels, J., Huysmans, M.: Vadose zone lag time effect on groundwater drought in a temperate climate. Water 12, 2123, doi:10.3390/w12082123, 2020.

Yevjevich, V.: An objective approach to definitions and investigations of continental hydrologic droughts, Hydrol. Pap. Color. State Univ. Fort Collins CO, 23, 25 pp., 1967.